# Specific Disruption of Ras2 CAAX Proteolysis Alters Its Localization and Function

Rajani Ravishankar,[a] Emily R. Hildebrandt,[a] Grace Greenway,[a] Nadeem Asad,[b] Sangram Gore,[b] Timothy M. Dore,[b,c] Walter K. Schmidt[a]

[a]Department of Biochemistry and Molecular Biology, University of Georgia, Athens, Georgia, USA
[b]New York University Abu Dhabi, Abu Dhabi, United Arab Emirates
[c]Department of Chemistry, University of Georgia, Athens, Georgia, USA

**ABSTRACT**   Many CAAX proteins, such as Ras GTPase, undergo a series of posttranslational modifications at their carboxyl terminus (i.e., cysteine prenylation, endoproteolysis of AAX, and carboxylmethylation). Some CAAX proteins, however, undergo prenylation-only modification, such as *Saccharomyces cerevisiae* Hsp40 Ydj1. We previously observed that altering the CAAX motif of Ydj1 from prenylation-only to canonical resulted in altered Ydj1 function and localization. Here, we investigated the effects of a reciprocal change that altered the well-characterized canonical CAAX motif of *S. cerevisiae* Ras2 to prenylation-only. We observed that the type of CAAX motif impacted Ras2 protein levels, localization, and function. Moreover, we observed that using a prenylation-only sequence to stage hyperactive Ras2-G19V as a farnesylated and nonproteolyzed intermediate resulted in a different phenotype relative to staging by a genetic *RCE1* deletion strategy that simultaneously affected many CAAX proteins. These findings suggested that a prenylation-only CAAX motif is useful for probing the specific impact of CAAX proteolysis on Ras2 under conditions where other CAAX proteins are normally modified. We propose that our strategy could be easily applied to a wide range of CAAX proteins for examining the specific impact of CAAX proteolysis on their functions.

**IMPORTANCE**   CAAX proteins are subject to multiple posttranslational modifications: cysteine prenylation, CAAX proteolysis, and carboxylmethylation. For investigations of CAAX proteolysis, this study took the novel approach of using a proteolysis-resistant CAAX sequence to stage *Saccharomyces cerevisiae* Ras2 GTPase in a farnesylated and nonproteolyzed state. Our approach specifically limited the effects of disrupting CAAX proteolysis to Ras2. This represented an improvement over previous methods where CAAX proteolysis was inhibited by gene knockout, small interfering RNA knockdown, or biochemical inhibition of the Rce1 CAAX protease, which can lead to pleiotropic and unclear attribution of effects due to the action of Rce1 on multiple CAAX proteins. Our approach yielded results that demonstrated specific impacts of CAAX proteolysis on the function, localization, and other properties of Ras2, highlighting the utility of this approach for investigating the impact of CAAX proteolysis in other protein contexts.

**KEYWORDS**   CAAX pathway, CAAX proteolysis, palmitoylation, Ras2, Rce1, shunt pathway, lipidation, proteolysis, Ras, Ras signaling, yeast

CAAX proteins are defined by a carboxyl-terminal CAAX motif: C (cysteine), A (an aliphatic amino acid), and X (one of several possible amino acids). The CAAX motif is often subject to three modifications: (i) isoprenylation at the cysteine by farnesyltransferase or geranylgeranyltransferase I; (ii) endoproteolytic removal of the AAX residues by Rce1 CAAX protease; (iii) carboxylmethylation by isoprenylcysteine carboxyl methyltransferase (ICMT) (Fig. 1). Whereas the prenyltransferases are cytosolic enzymes, Rce1 and ICMT are endoplasmic reticulum (ER) membrane-embedded enzymes (1–4). Some CAAX proteins

Address correspondence to Walter K. Schmidt, wschmidt@uga.edu.

The authors declare no conflict of interest.

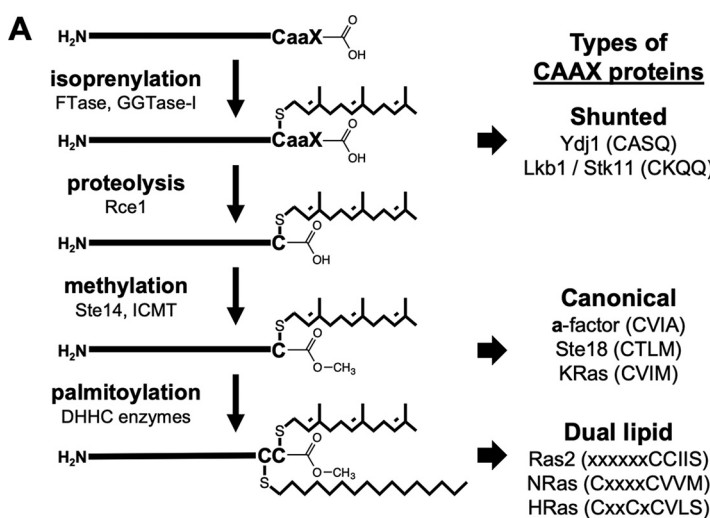

**FIG 1** Posttranslational modifications of CAAX proteins. (A) Initial isoprenylation of a consensus cysteine amino acid within the carboxyl-terminal CAAX motif yields a farnesylated ($C_{15}$; shown) or geranylgeranylated ($C_{20}$) protein. Shunted CAAX proteins are not further modified. Canonical CAAX proteins are subject to additional endoproteolysis and carboxylmethylation. Some canonical CAAX proteins, including certain Ras isoforms, are also modified with a palmitoyl lipid on a nearby cysteine (e.g., yeast Ras2, human NRas, and human HRas). CAAX proteolysis is not required for palmitoylation of mammalian HRas, and this issue was investigated for Ras2 in this study. (B) Ras2-CAAX variants evaluated in this study and their expected PTMs.

additionally undergo palmitoylation ($C_{16}$ lipidation) at a cysteine near the CAAX motif by DHHC palmitoyl acyltransferases, which are embedded in various intracellular membranes (5, 6). For example, *Saccharomyces cerevisiae* (budding yeast) Ras2 is palmitoylated by the ER-localized Erf2-Erf4 complex.

Recently, we reported that the budding yeast Hsp40 chaperone Ydj1 is an atypical CAAX protein that only undergoes prenylation and is not further modified by Rce1 and ICMT (7). The function and localization of Ydj1 depend on this alternative CAAX processing outcome that we refer to as the shunt pathway (Fig. 1). Growing evidence suggests that other CAAX proteins are shunted, but the importance of shunting in these instances remains to be investigated (8–11). The Ydj1 CASQ sequence retains its prenylation-only properties when transferred to yeast proteins that are otherwise canonically modified (i.e., **a**-factor mating pheromone and Ras2 GTPase) (7). Here, we take advantage of the CASQ sequence to examine the impact of altering the CAAX processing dynamics of budding yeast Ras2. Many well-described assays and phenotypes associated with Ras2 have been previously used to examine the effect of staging Ras2 in a prenylation-only state, typically through disruption of the *RCE1* CAAX protease gene. However, this approach simultaneously disrupts the processing dynamics of other CAAX proteins, leading to unclear interpretations with respect to specific impacts on Ras2-dependent function. We predicted that the prenylation-only CASQ sequence could be used as a new tool to reveal details about the specific impact of CAAX proteolysis toward the properties of Ras2 in an otherwise-normal prenylome background.

Ras GTPases regulate growth and division of eukaryotic cells and are critical to cell survival. In humans, there are multiple Ras isoforms, and hyperactive Ras mutants are associated with about 25% of human cancers (12). Hence, Ras has received much research and pharmacological attention. Across species and isoforms, Ras undergoes canonical CAAX modifications,

with some isoforms being additionally palmitoylated ($C_{16}$ lipidation) at a cysteine near the CAAX motif (13) (Fig. 1). In budding yeast, Ras1 and Ras2 are the two Ras isoforms, and at least one must be functional for cell viability. Both yeast Ras isoforms signal through the cAMP-protein kinase A pathway, with the Ras2 isoform having a more prominent role, especially during heat and nutrient stresses (13–15). In mammalian and yeast cells, CAAX modifications and palmitoylation regulate the association of Ras isoforms with the plasma membrane and internal membranes (e.g., ER, mitochondria, and Golgi), which are important sites of Ras activity (16–19).

In this study, we leveraged the shunted CASQ sequence for staging budding yeast Ras2 in a farnesylated and nonproteolyzed state to investigate the impact of CAAX cleavage on Ras2 properties through Ras-specific growth and localization assays. This approach focused the loss of CAAX proteolysis to Ras2 and avoided pleiotropic effects that would result from disrupting Rce1 CAAX protease activity through an *RCE1* deletion or chemical inhibition strategy. Overall, we observed that CAAX posttranslational modifications (PTMs) impacted several properties of Ras2 in addition to regulating its localization and that an important relationship exists between CAAX proteolysis and palmitoylation that is required for optimal Ras2 function. These findings suggest that shunted sequences could be broadly leveraged to investigate the impact of CAAX proteolysis on specific CAAX proteins under otherwise-normal prenylome conditions.

## RESULTS

**Ras2-CASQ supports yeast viability.** Yeast encodes two Ras paralogs (Ras1 and Ras2); one is minimally required for viability. While none of the enzymes that modify the Ras CAAX motif is essential for yeast viability, several assays have been developed to reveal the impact of these modifications on Ras2 activity. For example, fully modified Ras2-CIIS supports growth of *ras1Δ* yeast, but unmodified Ras2-SSIIS does not (17). (It should be noted that the carboxyl-terminal sequence of Ras2 ends with CCIIS. The first Cys is the known palmitoylation site. This site should be inferred as being present on the various sequences described in this study [e.g., CIIS] unless it has been mutated. For the latter, the mutated palmitoylation site will be explicitly reported [e.g., SCIIS]). To investigate the properties of farnesylated and nonproteolyzed Ras2, we examined the ability of Ras2-CASQ and other Ras2-CAAX variants to support growth in a viability assay (Fig. 2). Neither the vector control nor unmodified Ras2-SSIIS supported growth, consistent with previous findings that viability requires a form of modified Ras2 (17, 20). The ability of farnesylated Ras2 to support robust growth occurred whether it was cleaved (Ras2-CIIS), uncleaved (Ras2-CASQ), or nonpalmitoylated (Ras2-SCIIS). Decreased viability was observed, however, for farnesylated Ras2 that was both uncleaved and nonpalmitoylated (Ras2-SCASQ), which was more readily apparent through quantitative measures. These results indicated that yeast viability requires Ras2 farnesylation and either CAAX cleavage or palmitoylation as a secondary PTM and that combined loss of both secondary PTMs negatively impacts Ras2 function.

**Ras2-CASQ is farnesylated and palmitoylated.** To firmly attribute the reduced viability of yeast expressing Ras2-SCASQ to loss of CAAX cleavage, we examined whether Ras2-SCASQ might otherwise be underprenylated. Numerous CAAX proteins exhibit a mobility difference on SDS-PAGE because of farnesylation, and Ras also exhibits a partial mobility shift because of CAAX proteolysis (21–24). We thus performed mobility studies to establish the farnesylation status of each Ras2-CAAX variant being investigated, including those in which the palmitoylation site was altered.

Under the reported SDS-PAGE gel conditions, we observed differential mobility of fully modified Ras2-CIIS and unmodified Ras2-SSIIS (Fig. 3, top panels). The mobility of fully CAAX-modified but nonpalmitoylated Ras2-SCIIS was very similar to that of Ras2-CIIS, indicating that the bulk of the mobility shift is mainly attributable to CAAX PTMs and not palmitoylation. Uncleaved Ras2-CASQ and Ras2-SCASQ had mobility shifts that were intermediate, confirming that farnesylation and CAAX proteolysis independently contribute to the mobility shift of Ras2 (25). It did not escape our attention that protein level differences existed between variants, which is discussed in more detail below. Together, these data indicated that farnesylation of Ras2 is indifferent to whether the Ras2 palmitoylation site is present or the CAAX motif is cleaved. Of note, farnesylation of Ras2 is also indifferent to the presence of an

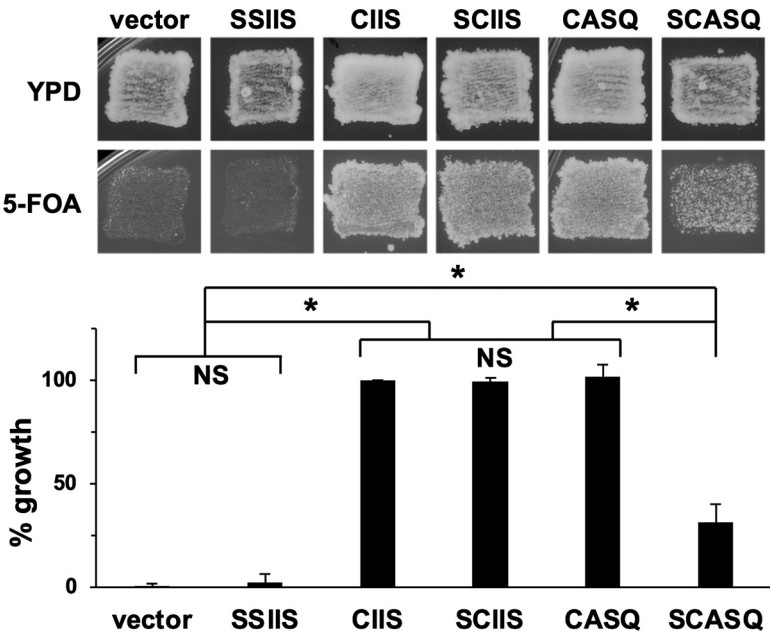

**FIG 2** Impact of CAAX motifs on Ras2 viability function. The 5-FOA plasmid-loss assay was performed with yeast initially coexpressing Ras1 from *URA3*-marked plasmid and the indicated Ras2-CAAX variant from a *LEU2*-marked plasmid. Growth on 5-FOA medium, indicative of the functional ability of the Ras2-CAAX variant, was assessed by qualitative patching (upper) or quantitative colony counting (bottom). For the qualitative test, cells were first patched on YPD solid media, on which cell growth could be supported by either Ras1 or a functional Ras2-CAAX variant, and then replica plated onto 5-FOA solid medium, on which cell growth could only be supported by a functional Ras2-CAAX variant. For quantitative assessment, strains cultured in SC-Ura, Leu were plated at low density on YPD solid medium to determine CFU, and resultant colonies were replica plated onto 5-FOA solid medium to determine the number of cells that could be supported by the indicated Ras2-CAAX variant. Percent growth was determined by deriving the ratio of colony counts on the 5-FOA to YPD plates for each strain and comparing the ratio to that of Ras2-CIIS, which was set to 100%. Values for the data shown in the graph were averaged from at least 4 independent experiments having both biological and technical replicates, and the error bars depict standard errors of the means. Statistical analysis was performed using one-way analysis of variance and Bonferroni's *post hoc* test. *, $P < 0.001$; NS, not significant. The yeast strain used was RJY510; plasmids used were pRS315, B250, Ras2-Scaax, Ras2-SsaaX, pWS1613, and pWS1615.

amino-terminal tag, based on prior mobility studies of green fluorescent protein (GFP)-tagged Ras2-CAAX variants (7).

While palmitoylation of different Ras isoforms is traditionally thought to require prior prenylation, palmitoylation has been detected in the absence of prenylation or full CAAX processing (21, 26–28). We directly investigated this issue for Ras2 using the acyl-polyethylene glycol (PEG) exchange (APE) assay, which has been applied for investigations of mammalian Ras palmitoylation (29). As expected, we observed palmitoylation of Ras2-CIIS, which was evident as a methoxypolyethylene glycol maleimide (mPEG-Mal)-dependent mobility shift in the presence of hydroxylamine (Fig. 3B). No shift was observed without hydroxylamine treatment. Also as expected, no shift was observed for Ras2-SCIIS lacking a palmitoylation site. For Ras2-CASQ, a shift was observed. Moreover, comparable shifts relative to total signal were observed for Ras2-CIIS and Ras2-CASQ. These results indicated that uncleaved Ras2-CASQ is subject to palmitoylation. Combined, these data suggest that CAAX proteolysis is not a prerequisite for Ras2 palmitoylation.

**The steady-state protein level of Ras2-CASQ is elevated relative to that of wild-type Ras2.** We considered that a trivial explanation for the reduced function of Ras2-SCASQ and Ras2-SSIIS in the viability assay could be reduced protein levels. The opposite was revealed, however, when we examined the steady-state protein levels of Ras2-CAAX variants (Fig. 4A and Fig. S2A in the supplemental material). Protein levels were generally higher for all Ras2-CAAX variants relative to wild-type Ras2-CIIS. Ras2-CASQ and Ras2-SCIIS levels were moderately higher (∼5× to 7×), and Ras2-SCASQ and Ras2-SSIIS levels were substantially higher (∼22×). We obtained similar results using cells harvested at

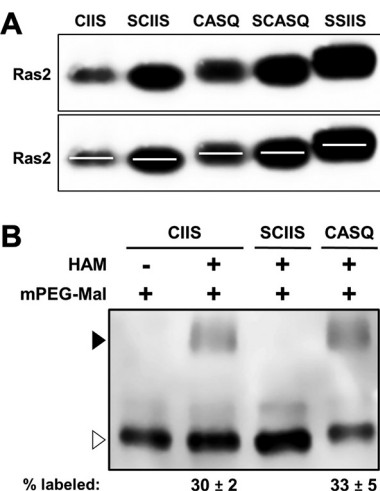

**FIG 3** Impacts of CAAX motifs on Ras2 farnesylation and palmitoylation. (A) Yeast cells expressing the indicated Ras2-CAAX variants were cultured in SC-Ura, Leu, and whole-cell lysates prepared by alkaline lysis were analyzed by SDS-PAGE (6% stacking; 10% resolving) and anti-Ras2 immunoblotting. A duplicate image of the immunoblot (lower panel) was modified with lines centered on each band for better visualization of mobility differences. Ponceau S staining was used to visualize loading across lanes (see Fig. S1 in the supplemental material). The strain used was RJY510; plasmids used were pRS315, B250, Ras2-Ssaax, Ras2-Scaax, pWS1613, and pWS1615. (B) The acyl-PEG exchange assay was performed on yeast cell lysates containing Ras2-CAAX variants. Equivalent amounts of each lysate were treated with (+) or without (−) hydroxylamine (HAM) followed by 5-kDa mPEG-Mal prior to evaluation by SDS-PAGE and immunoblotting with anti-Ras2 antibody. The mPEG-Mal-labeled and unlabeled populations are indicated (closed and open triangles, respectively). The values for percent labeled were derived by taking the ratio of labeled to total signal in each lane. The labeling pattern was reproducible across at least 2 independent experiments; a representative result is shown. The strain used was RJY510 transformed with plasmids B250, Ras2-Scaax, and pWS1613; transformants were selected on 5-FOA to express only the Ras2 isoform.

different densities (Fig. S2B), an alternative cell lysis method (Fig. S2C), and with cells from a different strain background (Fig. S2D). Overall, we consistently observed an inverse correlation between steady-state Ras2 levels and yeast viability phenotypes. Thus, we concluded that decreased expression could not be the root cause of the reduced abilities of Ras2-SCASQ and Ras2-SSIIS to support viability.

Previous studies have reported that CAAX PTMs influence the protein levels of certain CAAX proteins, potentially by affecting protein turnover (28, 30, 31). Based on these reports, we evaluated the turnover rate of Ras2-CAAX variants using cycloheximide chase analysis (Fig. 4B and Fig. S3). Kar2 was used as a control because it has a long half-life per proteomic analysis methods (32). Consistently, we estimated a half-life of >26 h for Kar2 with our cycloheximide-based method. Wild-type Ras2-CIIS had an estimated half-life of 1.6 h, which was the shortest half-life of the Ras2-CAAX variants evaluated. Ras2-CASQ had a slightly longer half-life of 1.9 h, and Ras2-SCIIS had an even longer half-life of 5.4 h. By comparison, Ras2-SCASQ and Ras2-SSIIS had half-lives of 26 h or more. The slower turnover rates for Ras2-CAAX variants roughly correlated with their observed increases in steady-state protein levels in whole-cell lysates (Fig. 3A and 4A).

**The level of GTP-activated Ras2-CASQ is elevated relative to wild-type Ras2.** As part of its normal GTPase function, Ras2 receives an upstream signal, exchanges GDP for GTP to become activated, and then hydrolyzes GTP to GDP to become inactive until the start of another functional cycle (13). Interactions with effectors occur with the activated state. Because protein levels did not readily explain the decreased viability of yeast expressing Ras2-SCASQ and Ras2-SSIIS, we investigated whether GTP activation was reduced for these variants. We evaluated the GTP activation of Ras2-CAAX variants using a Ras pulldown activation assay, a routine technique that measures the binding of activated Ras-GTP to the Ras-binding-domain (RBD) of the mammalian effector Raf (33, 34).

We determined the RBD-bound fraction of each Ras2-CAAX variant (i.e., Ras2-GTP) relative to total Ras2-CAAX in the sample, and we compared this ratio with that of fully

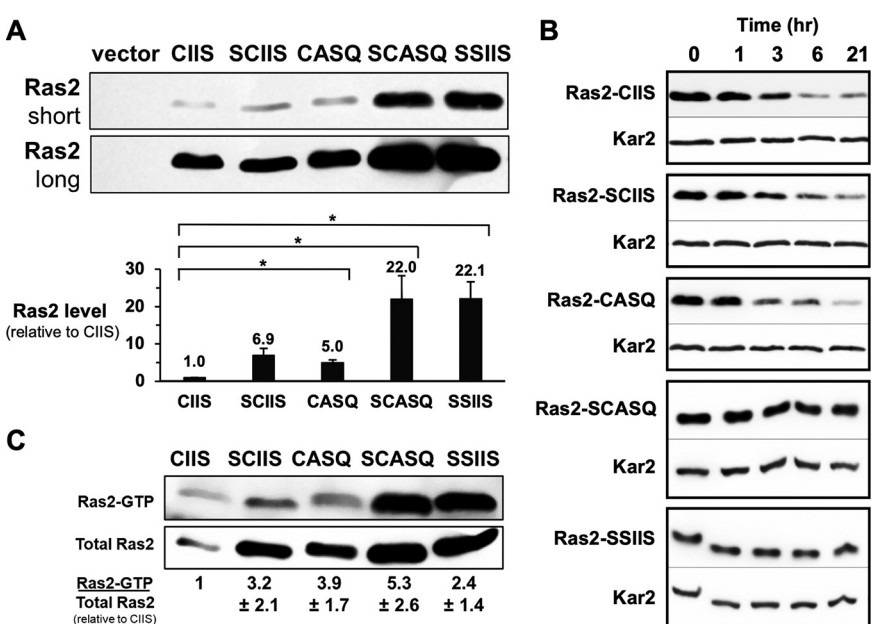

**FIG 4** Impacts of CAAX motifs on Ras2 protein levels and GTP loading. (A) Yeast cells grown in SC-Ura, Leu liquid medium for setting up the 5-FOA plasmid-loss assay (see Fig. 2) were used in parallel for whole-cell lysate preparation by alkaline lysis. Short- and long-time exposures of the anti-Ras2 immunoblot are shown. Equivalent loading was assessed by Ponceau S staining (see Fig. S2A in the supplemental material). The strain used was RJY510; plasmids used were pRS315, B250, Ras2-Scaax, Ras2-Ssaax, pWS1613, and pWS1615. For quantification, immunoblot band intensities were evaluated using ImageJ from at least 4 independent experiments, inclusive of panel A data, that reflected different cell growth and extract preparation conditions (see Fig. S2B to D). (B) Late-log-phase cultures of yeast cells grown in SC-Leu liquid medium were treated with 250 $\mu$g/mL cycloheximide ($t$ = 0 min), and samples were collected at the indicated time points for whole-cell lysate preparation by alkaline lysis. Samples were analyzed by anti-Ras2 and anti-Kar2 immunoblotting. The strain used was RJY510; plasmids used were B250, Ras2-Scaax, Ras2-Ssaax, pWS1613, pWS1615, and pWS1890; transformants were selected on 5-FOA to only express the Ras2 isoform, except for the transformant expressing Ras2-SSIIS, which only grows when Ras1 is coexpressed. (C) The Ras pulldown activation assay was performed using a commercial kit on cell lysates prepared by bead-beating of cells. Equivalent amounts of each lysate were used for the assessment of input Ras2 (total Ras2) and activated Ras2 (Ras2-GTP), which were confirmed by Ponceau S staining (see Fig. S2E). Ratios of Ras2-GTP to total Ras2 relative to Ras2-CIIS were calculated for multiple exposures across 2 experiments, except for SSIIS, whose ratio was calculated for multiple exposures from a single experiment. Average ratios are indicated below each lane; error bars depict standard deviations. The strains used were as described for panel B.

modified Ras2-CIIS (Fig. 4C and Fig. S2E). Fully modified Ras2-CIIS had the lowest GTP activation ratio, Ras2-SCIIS and Ras2-CASQ, which lack 1 PTM, had intermediate ratios of activation, and Ras2-SCASQ, which lacks 2 PTMs, had the highest GTP activation ratios. Unmodified Ras2-SSIIS also had intermediate GTP activation. These findings indicated that GTP activation does not depend on CAAX modifications and that the inability of Ras2-SCASQ and Ras2-SSIIS to fully support viability is not due to low protein or GTP activation levels.

**Both CAAX cleavage and palmitoylation are required for plasma membrane localization of Ras2.** Because Ras2-SCASQ had reduced function despite elevated levels of protein and GTP activation, we considered whether altered localization could explain its diminished signaling. We thus performed differential fractionation to evaluate the extent that each Ras2-CAAX variant was membrane associated (Fig. 5A). This analysis revealed that Ras2-CIIS variants with and without palmitoylation were enriched in the particulate fraction (>90%), indicative of membrane association, as previously observed (35). Uncleaved Ras2-CASQ was also enriched in the particulate fraction but less so (~75%), while uncleaved and nonpalmitoylated Ras2-SCASQ and unmodified Ras2-SSIIS were significantly de-enriched in the particulate fraction (~15%). Generally, more PTMs correlated with more enrichment in the particulate fraction. We also performed localization studies with GFP-Ras2-CAAX variants, where the GFP tag was attached to the amino terminus so as not to interfere with the

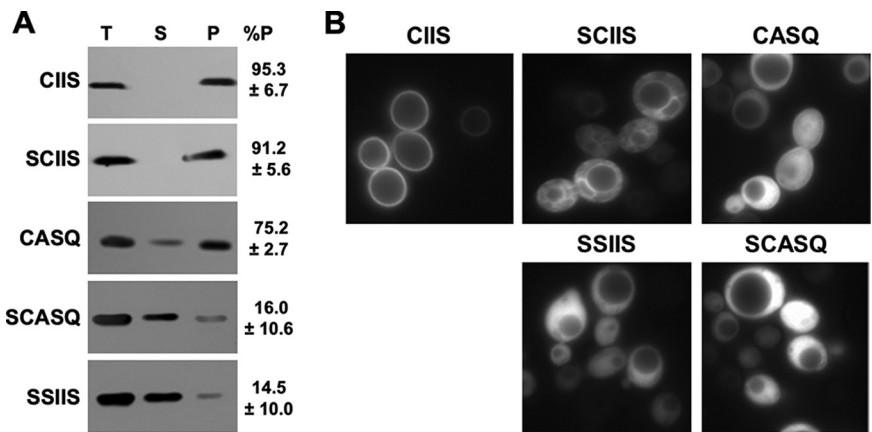

**FIG 5** Impact of CAAX motifs on Ras2 localization. (A) Total cell lysates of strains expressing Ras2-CAAX variants were prepared by bead-beating and used for differential fractionation. Equivalent percentages of each sample representing the total lysate (T), supernatant (S), and particulate (P) fractions from 100,000 × g centrifugation were evaluated by immunoblotting with anti-Ras2 antibody. The percentage of signal associated with the P fraction relative to the sum of signal from P and S fractions is noted, and values were obtained from at least 2 independent experiments. The strain used was RJY510; plasmids used were B250, Ras2-Scaax, Ras2-Ssaax, pWS1613, and pWS1615. (B) Fluorescence microscopy was performed with the GFP-tagged Ras2-CAAX variants under fluorescence optics (63×). The image shown for each variant is representative of at least 30 fields of cells. The strain used was LRB938; plasmids used were pWS1714, pWS1735, pWS1821, pWS1823, and pWS1889.

carboxyl-terminal PTMs. In agreement with our fractionation studies and previous reporting, fully modified GFP-Ras2-CIIS was localized at the plasma membrane, nonpalmitoylated GFP-Ras2-SCIIS at internal membranes, and unmodified GFP-Ras2-SSIIS in the cytosol (Fig. 5B) (17). GFP-Ras2-CASQ had cytosolic localization; the population identified to be membrane-associated by fractionation was not visually evident, which could have been due to this pool being eclipsed by the cytosolic population or a consequence of the GFP tag impairing membrane association, as has been reported elsewhere (35). GFP-Ras2-SCASQ displayed primarily cytosolic localization, in strong agreement with fractionation studies. The protein levels of GFP-Ras2-CAAX variants were not as variable as observed for untagged Ras2 variants (Fig. S4A). This could be attributed to use of the strong *YDJ1* promoter to express GFP-Ras2 or the GFP tag interfering with normal Ras2 protein turnover. Considering both fractionation and GFP-tagging results, it is clear that Ras2-SCASQ and Ras2-SSIIS were mislocalized, which could potentially explain their decreased functional abilities.

Because of the unknown impact of the GFP tag on Ras2 function, we additionally confirmed that the GFP-tagged Ras2 variants functioned similarly to their untagged counterparts. For this purpose, we used a Ras function assay that accommodated the *URA3*-marked GFP-Ras2 constructs that could not be tested using the 5-FOA plasmid-loss assay. This approach also offered the ability to test the function of Ras2-CAAX variants in an orthogonal assay, in which growth reflects the ability of variants to complement a *ras2-23* temperature sensitive (TS) mutation (36). The growth phenotype trends observed with the *ras2-23*(TS) assay (Fig. S4B) paralleled those observed with the 5-FOA plasmid-loss assay (Fig. 2), especially for Ras2-SCASQ and Ras2-SSIIS which were less capable of supporting growth in both assays.

**Ras2-G19V-CASQ exhibits signaling effects intermediate to fully modified and unmodified Ras2-G19V-CAAX variants.** Hyperactive mutations of Ras affect cell growth in mammalian cells, where such mutations are commonly associated with cancer, and in yeast, where mutations have a variety of well-documented effects, such as sensitizing yeast to acute heat shock (17). The heat shock assay takes advantage of the sensitization of stationary-phase yeast to acute heat shock and was used to compare the phenotypes of yeast expressing Ras2-G19V-CAAX variants. In the assay, two doses of *RAS2* are present; one is chromosomally encoded Ras2-CIIS, and the other is the plasmid-encoded Ras2-CAAX variant being evaluated. Consistent with previous reporting, yeast cells expressing wild-type Ras2-CIIS were insensitive to acute heat shock

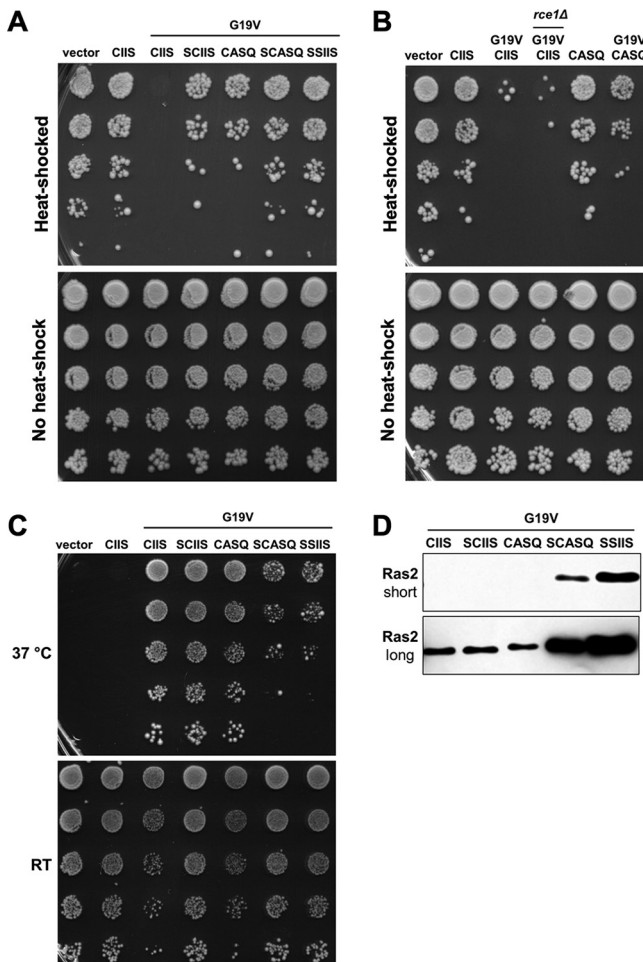

**FIG 6** Impact of CAAX motifs on Ras2-G19V activity. (A and B) The acute heat shock assay was performed using yeast expressing the indicated Ras2-CAAX variants. Saturated cultures grown in SC-Leu medium were normalized for density, heated (55°C, 10 min) or mock treated (no heat shock), and serially diluted 1:5 before pinning on SC-Leu (heat-shocked) and YPD (no heat shock) plates. Yeast strains used were LRB938 and yWS3126; plasmids used were pRS315, B250, B561, B562, pWS1612, pWS1614, and pWS1890. (C) The *cdc25*(TS) temperature sensitivity assay was performed using yeast transformed with Ras2-CAAX variants. The strains were cultured in SC-Leu liquid medium, normalized for density, serially diluted 1:10, and pinned on two SC-Leu plates that were incubated at permissive room temperature (RT) and restrictive temperature (37°C) up to 5 days. The yeast strain used was *cdc25-2*; plasmids used were pRS315, B250, B561, B562, pWS1612, pWS1614, and pWS1890. For panels A to C, the data shown are representative of the patterns of growth observed across at least 3 independent experiments having both biological and technical replicates; representative results are shown. (D) Whole-cell lysates were prepared by alkaline lysis and analyzed as described for Fig. 4A. Equivalent portions of each lysate were evaluated, except for Ras2-G19V-CIIS, which was evaluated in a higher amount to facilitate immunoblot detection (see Fig. S1). Short- and long time-exposures of the anti-Ras2 immunoblot are shown. The strain used was RJY510; plasmids used were B561, B562, pWS1612, pWS1614, and pWS1890.

(55°C, 10 min), which was indicated as outgrowth post-heat shock (Fig. 6A, upper panel) (17). This occurred whether the Ras2-CIIS dosage was single (i.e., vector) or double (i.e., CIIS). By comparison, yeast cells producing hyperactive Ras2-G19V-CIIS were highly sensitive to heat shock and displayed extremely limited outgrowth, consistent with Ras2-G19V-CIIS being a dominant-negative mutant. We also confirmed previous reporting that Ras2-G19V-SSIIS and Ras2-G19V-SCIIS mitigated heat shock sensitivity, and we expanded this observation to Ras2-G19V-CASQ and Ras2-G19V-SCASQ (17, 20). Desensitization was observed for Ras2-G19V-CASQ, indicating that it has lower activity than Ras2-G19V-CIIS. To be sure that observed differences were not due to differential cell inputs, mock-treated samples (i.e., no heat) were processed in parallel and pinned onto nonselective yeast extract-peptone-dextrose (YPD) (Fig. 6A, lower panel). Overall,

the heat shock assay demonstrated that loss of any PTM on Ras2 reduced the signaling effects of the Ras2-G19V mutant.

Prior studies have demonstrated that *rce1Δ* yeast are partially desensitized to the heat shock effect associated with Ras2-G19V (36). However, it remains unclear whether desensitization is specifically due to uncleaved Ras2-G19V-CIIS or the ensemble of uncleaved CAAX proteins arising from *RCE1* deletion. Thus, we compared the heat shock recovery of *rce1Δ* yeast cells expressing Ras2-G19V-CIIS with that of wild-type yeast expressing Ras2-G19V-CASQ. We observed poor growth for *rce1Δ* and significantly better growth for Ras2-G19V-CASQ, suggesting that the *RCE1* deletion may be negatively impacting additional signaling pathways needed for survival of heat shock other than Ras2 (Fig. 6B and Fig. S5A).

We further analyzed the activity of Ras2-G19V-CAAX variants in an orthogonal assay that monitored the ability of hyperactive Ras2 to rescue the temperature sensitivity of *cdc25* (TS) yeast. Cdc25 is a Ras2 guanine nucleotide exchange factor (GEF) required for GTP activation of Ras2. The temperature sensitivity of *cdc25*(TS) yeast can be bypassed with constitutively GTP-activated Ras (37, 38, 39). The *cdc25*(TS) temperature sensitivity test revealed that multiple PTMs had to be lost before Ras2 function was compromised. Uncleaved Ras2-G19V-CASQ and nonpalmitoylated Ras2-G19V-SCIIS each rescued growth equivalent to fully modified Ras2-G19V-CIIS, but uncleaved and nonpalmitoylated Ras2-G19V-SCASQ only partially rescued growth (Fig. 6C). This trend mirrored that observed for wild-type Ras2-CAAX variants in the 5-FOA plasmid-loss viability assay and *ras2-23*(TS) temperature sensitivity assay, indicating a similar impact of Ras2 PTMs on yeast viability with either Ras2 or Ras2-G19V status (Fig. 2 and Fig. S4B). Taken together, these results indicate that Ras2-G19V-CAAX variants lacking either cleavage or palmitoylation each have sufficient function to rescue growth of *cdc25*(TS) yeast but not function so elevated to have detrimental signaling effects like fully modified Ras2-G19V-CIIS in the heat shock assay. Additionally, we observed that protein levels were increased for Ras2-G19V-CAAX variants with fewer PTMs (Fig. 6D and Fig. S5B). Thus, PTMs can modulate both the activity and protein level of both wild-type and hyperactive Ras2.

## DISCUSSION

This work primarily investigated the role of CAAX proteolysis in regulating the properties of Ras2. Previous work on this topic has typically disrupted the *RCE1* CAAX protease gene or chemically inhibited the Rce1 enzyme, potentially leading to simultaneous impacts on many CAAX proteins. A lack of good methodology to circumvent this problem has kept the field from understanding the specific effects of inhibiting CAAX proteolysis on any individual CAAX protein. Here, we have described a novel method that uses the uncleaved CAAX sequence CASQ to stage budding yeast Ras2 as a farnesylated and uncleaved intermediate (i.e., "shunted" Ras). Our approach has allowed for a critical examination of the impact of CAAX proteolysis on Ras2 function. Our results clearly indicate that CAAX proteolysis and palmitoylation are somewhat interchangeable for Ras2 activity yet provide distinct functionality and properties to Ras2. Moreover, we determined with shunted Ras2-CASQ that CAAX proteolysis is not prerequisite for Ras2 palmitoylation, which agrees with findings for mammalian Ras (21). The implied order of PTMs illustrated in Fig. 1 is thus an oversimplification for palmitoylation of Ras proteins and perhaps other dually lipidated CAAX proteins.

We also used shunted Ras2-CASQ to compare the importance of CAAX proteolysis with other carboxyl-terminal PTMs for Ras2 function. We observed that Ras2 requires two or more PTMs—farnesylation and palmitoylation or farnesylation and proteolysis—for Ras2-dependent viability (i.e., Ras2-SSIIS < Ras2-SCASQ < Ras2-CASQ ≈ Ras2-SCIIS ≈ Ras2-CIIS) (Fig. 2). The pattern of function largely mirrored that observed for Ras2 membrane localization (i.e., Ras2-SSIIS ≈ Ras2-SCASQ < Ras2-CASQ < Ras2-SCIIS < Ras2-CIIS), as determined by Ras2 fractionation and GFP-Ras2 microscopy. We interpreted our localization results to indicate that the membrane association of Ras2 improves with PTMs in an additive fashion. Our localization studies further suggested that nonproteolyzed Ras2-CASQ is localized more on internal

membranes and in the cytosol than on plasma membrane (PM), in agreement with previous findings examining the impact of an *RCE1* deletion on Ras2 localization (36, 40). Traditionally, Ras is reported to signal primarily from the PM, with CAAX PTMs and palmitoylation being important for effective PM association. Yet, when proteolysis does not occur, Ras2 delocalized from the PM can still support yeast viability, suggesting that Ras2 signaling can emanate from internal membranes. Similar observations have been made for Ras2 delocalized due to defects in palmitoylation (20). Indeed, Ras signaling from internal membranes is not without precedent, and yeast Ras2 regulators and effectors localize to internal membranes in addition to the PM (16, 41, 42).

To further understand how PTMs impact the properties of Ras2, we investigated whether the reduced function of Ras2-CAAX variants with fewer PTMs (i.e., Ras2-SCASQ and Ras2-SSIIS) was simply due to decreased protein levels or negative impacts on GTP activation. Instead, we generally observed an inverse relationship between the extent of PTMs and these properties of Ras2. Our results revealed higher steady-state protein levels for Ras2-CAAX variants with fewer PTMs, which we attributed to their slower protein turnover rates. While the mechanism of Ras2 turnover is unknown, mammalian Ras isoforms are subject to ubiquitin-mediated regulation and degradation (43, 44). In addition, others have reported that deficiency of Rce1 activity (i.e., CAAX proteolysis) increases steady-state levels of HRas and prelamin A, consistent with our conclusion that CAAX PTMs impact protein levels; a decrease in the prelamin A turnover rate has also been reported (30, 31). We also observed increased GTP activation for Ras2-CAAX variants with fewer PTMs. Similar to our results, both increased levels of protein and GTP activation occurred with loss of CAAX PTMs for *Schizosaccharomyces pombe* Rho2 (23). Clearly, more work will be needed to understand the inverse relationship of PTMs with Ras2 protein levels and GTP activation. An intriguing possibility is that cells respond with compensatory changes in protein turnover, transcription, translation, and/or GTP activation of Ras2 to achieve a threshold amount of protein and GTP activation needed for optimal function from lower-functioning variants.

For Ras2 and other CAAX proteins, CAAX PTMs are known to regulate both their membrane association and interactions with other proteins (17, 45–49). Indeed, we observed that Ras2-CAAX variants had altered localization (e.g., Ras2-CASQ) (Fig. 7A). Yet, these variants were GTP-loaded as well as if not better than fully modified Ras2-CIIS, suggesting that interactions with the Ras2 GEF are not negatively impacted by Ras2 PTMs. Consistent with our observations, loss of CAAX PTMs do not impact GTP activation of mammalian Ras or *Aspergillus fumigatus* RasA (28, 50). Because the different functional properties of Ras2-CAAX variants cannot be attributed to poor GTP loading (i.e., defective upstream events), we speculate that differences reflect negative impacts on downstream events, specifically effector engagement. In our model, we propose that the cell compensates with increased levels of total and activated Ras2 to enhance overall effector engagement when the effector engagement of individual Ras2 molecules is compromised due to incomplete PTM (Fig. 7B). This compensation increases total signaling intensity to achieve the threshold level needed for cell survival in some cases (i.e., Ras2-SCIIS and Ras2-CASQ) but not others (i.e., Ras2-SCASQ and Ras2-SSIIS). In this model, a key parameter appears to be the number of PTMs, with two being minimally needed for Ras2 to support the basal activity needed for viability signaling (i.e., growth in the 5-FOA plasmid-loss assay). Fully modified Ras2-CIIS with three PTMs is PM localized and has the highest signaling intensity due to high effector engagement. Ras2 with two PTMs (i.e., Ras2-CASQ and Ras2-SCIIS) is mislocalized to internal membranes and has less signaling intensity on a per-molecule basis relative to Ras2-CIIS, but increased protein levels compensate and yield a viability phenotype that is indistinguishable from that of fully modified Ras2-CIIS. For Ras2-CASQ, the two PTMs are farnesylation and palmitoylation, and for Ras2-SCIIS, they are farnesylation and proteolysis. By comparison, Ras2-SCASQ with its single PTM (i.e., farnesylation) is even more mislocalized and has weaker signaling intensity. Unmodified Ras2-SSIIS is mislocalized to a similar extent as Ras2-SCASQ but is even more impaired for signaling intensity. Neither Ras2-SCASQ nor Ras2-SSIIS can support normal signaling for viability despite their relatively high protein levels and GTP activation. The

**A**

| CAAX | Membrane association | Protein level | GTP activation | Viability |
|---|---|---|---|---|
| CIIS | +++ | + | + | +++ |
| SCIIS | +++ | ++ | ++ | +++ |
| CASQ | ++ | ++ | ++ | +++ |
| SCASQ | + | +++ | +++ | ++ |
| SSIIS | + | +++ | ++ | + |

**B**

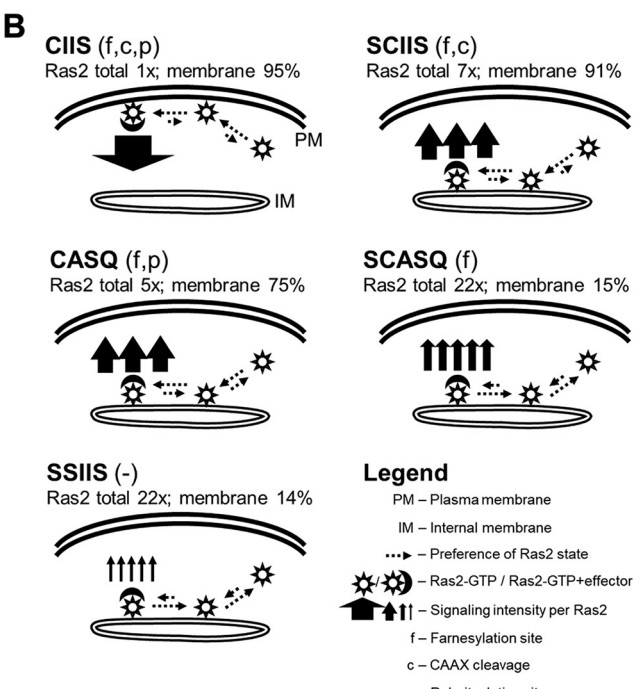

**CIIS** (f,c,p)
Ras2 total 1x; membrane 95%

**SCIIS** (f,c)
Ras2 total 7x; membrane 91%

**CASQ** (f,p)
Ras2 total 5x; membrane 75%

**SCASQ** (f)
Ras2 total 22x; membrane 15%

**SSIIS** (-)
Ras2 total 22x; membrane 14%

**Legend**
PM – Plasma membrane
IM – Internal membrane
┅▶ – Preference of Ras2 state
☼/☼ – Ras2-GTP / Ras2-GTP+effector
▲▲↑ – Signaling intensity per Ras2
f – Farnesylation site
c – CAAX cleavage
p – Palmitoylation site

**FIG 7** Model relating PTMs to properties of Ras2. (A) Qualitative representations of phenotypes for viability function (see Fig. 2), protein levels (total and GTP-activated) (see Fig. 4A and C, respectively), and Ras2 localization (see Fig. 5) observed for Ras2-CAAX variants examined in this study. The scaling is relative to the Ras2-CAAX variant, with the weakest phenotype observed within the set for each assay. (B) Our experimental observations were used to model the observed effects of PTMs. The model proposes that all Ras2-CAAX variants are GTP-activated but have different functional abilities due to varying effector engagements. Fully modified Ras2-CIIS (i.e., farnesylated, cleaved, and palmitoylated) is at the plasma membrane, has the lowest protein levels, and has highest signaling intensity; the widths of solid arrow(s) emanating from the membrane-bound Ras2/effector complex are indicative of intensity; the length and direction of dashed arrows represent the likelihood of a preferred state of membrane association or effector engagement. Partially modified Ras2-SCIIS and Ras2-CASQ (i.e., farnesylated and either CAAX cleaved or palmitoylated, respectively) are mislocalized from PM, have increased protein levels, and have reduced signaling intensity. Farnesylated-only Ras2-SCASQ and unmodified Ras2-SSIIS are mislocalized into the cytosol, have the highest protein levels, and have the lowest signaling intensities. We propose that increased Ras2 levels (total and GTP-activated) are a compensatory cellular response to increased overall Ras2 signaling output when loss of PTMs leads to reduced Ras2 signaling; the number of solid arrows is representative of this effect but not proportional to actual ratios.

function of each Ras2-CAAX variant thus reflects its level of effector engagement, which is a direct consequence of how well each variant associates with membranes to access effectors and/or can stably interact with effectors.

At present, our model does not explain why loss of a single PTM decreased signaling effects of hyperactive Ras2-G19V in the acute heat shock assay but not of wild-type Ras2 in the 5-FOA plasmid-loss assay. We reasoned that this could reflect differences in the signaling thresholds required for the two phenotypes. But, it is also possible that the two phenotypes result from distinct signaling pathways involving different Ras2 interactors. Our model also does not explain why Ras2-G19V-CASQ exhibits decreased

**TABLE 1** Yeast strains used in the study

| Strain | Genotype | Source |
|---|---|---|
| JRY5318 | *MAT**a** ade2-1 leu2-3,112 trp1 ura3-1 ras1Δ::HIS3 ras2-23(TS)* | 36 |
| LRB938 | *MAT**a** his3 leu2 ura3-52* | 17 |
| RJY510 | *MATα ras1::HIS3 ras2Δ ura3 his3 leu2 (YCp50 RAS1 URA3)* | 17 |
| cdc25-2 | *MATα ura3 lys2 leu2 trp1 hisΔ200 ade2-101 cdc25-2* | 37 |
| yWS2762 | *MAT**a** his3 leu2 ura3-52 ste24::KAN$^R$* | This study |
| yWS2802 | *MAT**a** his3 leu2 ura3-52 rce1::NAT$^R$ ste24::KAN$^R$* | This study |
| yWS2981 | *MATα his3 leu2 ura3-52 rce1::NAT$^R$ ste24::KAN$^R$* | This study |
| yWS3126 | *MAT**a** his3 leu2 ura3-52 rce1::NAT$^R$* | This study |

signaling in the yeast system while mammalian KRas-G12D exhibits increased signaling in a $Rce1^{-/-}$ murine hematopoietic cell system (51). These contrasting results could be system specific or a consequence of using an *RCE1* deletion approach that impacts multiple proteins and pathways (52). For mammalian Ras studies involving an *RCE1* deletion approach, Ras-CASQ variants could be used to obtain results by an orthogonal approach.

Overall, this work demonstrates that it is possible to stage and evaluate specifically Ras2 in a farnesylated and nonproteolyzed state (i.e., shunted) without resorting to approaches that broadly impact multiple CAAX proteins. Our approach revealed that shunted Ras2-CASQ retained the basal activity needed to support yeast viability but reduced the function of hyperactive Ras2-G19V. Additionally, our work indicated that either CAAX proteolysis or palmitoylation is needed for optimal Ras2 function. Future research of shunted Ras CAAX variants should allow further dissection of the roles of CAAX proteolysis and palmitoylation for Ras localization, protein levels, GDP-GTP cycling, and effector interactions. Finally, these studies support the potential utility of the CASQ sequence for probing the importance of CAAX proteolysis in other protein contexts, such as Rho, lamins, and the double-stranded RNA sensor Oas1 that is implicated in coronavirus disease 2019 risk (30, 53–56).

## MATERIALS AND METHODS

**Chemicals and reagents.** Unless otherwise noted, all fine chemicals were purchased from Sigma-Aldrich. Cycloheximide (Research Products International catalog number C81040) was prepared as a 25-mg/mL stock in ethanol. All restriction enzymes and T4 DNA ligase were purchased from New England Biolabs. All oligonucleotides used for generating PCR products or introducing mutations were purchased from Integrated DNA Technologies.

**Yeast strains, plasmids, and media.** Yeast strains, plasmids, and oligonucleotides used in this study are listed in Tables 1 and 2 and Table S1, respectively. Yeast strains were routinely propagated on YPD or selective solid yeast media at 30°C unless mentioned otherwise. Introduction of plasmids or plasmid fragments into yeast used a lithium acetate-based procedure followed by selection of transformants on appropriate selective solid media (7).

Several yeast strains were created for this study. yWS2762 (*ste24::KAN$^R$*) was created by replacing the *STE24* open reading frame in LRB938 with the kanamycin resistance gene (*KAN$^R$*). pWS405 was used as the source of *KAN$^R$*, as it is flanked by untranslated regions of the *STE24* locus that serve as sites for homologous recombination. yWS2802 (*ste24::KAN$^R$ rce1::NAT$^R$*) was created similarly by replacing the *RCE1* open reading frame in yWS2762 by recombination with the nourseothricin N-acetyltransferase gene (*NAT$^R$*). pWS714 was used as the source of *NAT$^R$* as it is flanked by untranslated regions of the *RCE1* locus that serve as sites for homologous recombination. Candidates recovered after appropriate selection on YPD solid media containing antibiotic (200 $\mu$g/mL G418 or 100 $\mu$g/mL nourseothricin) were evaluated using PCR to confirm the presence of the desired disruption. yWS3126 (*rce1::NAT$^R$*) was created by cross using LRB938 transformed with pRS316 and yWS2981 (*MATα* version of yWS2802), followed by random sporulation, screening of haploid candidates for mating type and desired genetic markers and gene disruptions, and propagation on 5-FOA solid medium (0.1% [wt/vol] 5-FOA [Research Products International]) to lose the pRS316 (*URA3*) plasmid.

Multiple plasmids were created for this study. Construction details are provided in Table S2. The sequences of all engineered plasmids were confirmed by Sanger sequencing for the open reading frames and 5′ and 3′ untranslated regions of encoded genes (Eurofins or Genewiz). Standard ligation or yeast homologous recombination methods were used to create plasmids (7).

**Cell lysis, SDS-PAGE, and immunoblotting.** For alkaline lysis, whole-cell lysates were prepared as previously described (57). Briefly, cells were treated with NaOH and $\beta$-mercaptoethanol, proteins were precipitated with trichloro acetic acid, and the protein precipitates were resuspended in urea sample buffer (250 mM Tris [pH 8.0], 6 M urea, 5% $\beta$-mercaptoethanol, 4% SDS, 0.01% bromophenol blue). Lysates were heated (65°C, 10 min) and clarified of insoluble material by centrifugation (16,000 × *g*, 1 min) prior to analyses.

For bead-beating lysis, cells were broken in lysis buffer (50 mM Tris-HCl [pH 7.5], 150 mM NaCl, 1 mM EDTA, 5 mM MgCl$_2$, 0.5% NP40, 0.25% sodium deoxycholate, with protease inhibitors 1 mM phenylmethylsulfonyl

**TABLE 2** Plasmids used in the study

| Plasmid | Genotype | Source |
|---|---|---|
| B250 | *CEN LEU2 RAS2-CIIS* | 35 |
| B561 | *CEN LEU2 RAS2-G19V-CIIS* | 35 |
| B562 | *CEN LEU2 RAS2-G19V-SCIIS* | 35 |
| p4339 | *NAT*[R] | 60 |
| pRS315 | *CEN LEU2* | 61 |
| pRS316 | *CEN URA3* | 61 |
| pSM1275 | *CEN URA3 RCE1* | 1 |
| pWS405 | *CEN URA3 ste24::KAN*[R] | 62 |
| pWS714 | *CEN URA3 rce1::NAT*[R] | This study |
| pWS1389 | *CEN URA3 P$_{YDJ1}$-GFP-YDJ1* | 7 |
| pWS1390 | *CEN URA3 P$_{YDJ1}$-GFP-YDJ1-SASQ* | 7 |
| pWS1501 | *CEN URA3 P$_{YDJ1}$-GFP* | This study |
| pWS1612 | *CEN LEU2 RAS2-G19V-CASQ* | This study |
| pWS1613 | *CEN LEU2 RAS2-CASQ* | This study |
| pWS1614 | *CEN LEU2 RAS2-G19V-SCASQ* | This study |
| pWS1615 | *CEN LEU2 RAS2-SCASQ* | This study |
| pWS1714 | *CEN URA3 P$_{YDJ1}$-GFP-RAS2-CASQ* | This study |
| pWS1735 | *CEN URA3 P$_{YDJ1}$-GFP-RAS2-CIIS* | This study |
| pWS1821 | *CEN URA3 P$_{YDJ1}$-GFP-RAS2-SCASQ* | This study |
| pWS1823 | *CEN URA3 P$_{YDJ1}$-GFP-RAS2-SCIIS* | This study |
| pWS1889 | *CEN URA3 P$_{YDJ1}$-GFP-RAS2-SSIIS* | This study |
| pWS1890 | *CEN LEU2 RAS2-G19V-SSIIS* | This study |
| Ras2-Ssaax | *CEN LEU2 RAS2-SSIIS* | 17 |
| Ras2-Scaax | *CEN LEU2 RAS2-SCIIS* | 17 |

fluoride, and 1 $\mu$g/mL each of aprotinin, chymostatin, leupeptin, and pepstatin) with 3 to 4 cycles of vortexing (30 to 40 s) and ice incubation (~60 s). The resulting lysate was centrifuged (13,000 × *g*, 5 min), and the clarified lysate protein concentration was quantified using the DC protein assay (Bio-Rad), mixed with 6× Laemmli sample buffer (375 mM Tris-HCl [pH 6.8], 10% SDS, 50% glycerol, 0.4% bromophenol blue, 10% $\beta$-mercaptoethanol), and heated (95℃, 5 min) prior to analyses.

SDS-PAGE (6% stacking gel and 12.5% resolving gel, unless otherwise mentioned) and immunoblotting were performed as described previously (7). Briefly, proteins were transferred to a nitrocellulose membrane and then blocked with Tris-buffered saline with Tween 20 (TBST; 100 mM Tris [pH 7.5], 400 mM NaCl, 0.1% Tween 20) containing 5% milk. The blots were sequentially incubated with appropriate primary and horserad-ish peroxidase (HRP)-conjugated secondary antibodies, washed with TBST and Tris-buffered saline (TBS), treated with enhanced chemiluminescence (ECL) reagent (Advansta WesternBright ECL spray or Kindle Biosciences KwikQuant Western blot detection kit [catalog number R1004]), and chemiluminescence was detected using a KwikQuant Imager system (Kindle Biosciences). Digital images of the immunoblots were subjected to minor cropping and contrast adjustment using Adobe Photoshop or ImageJ; protein bands were quantified using ImageJ. Antibodies were diluted into TBST containing 1% milk unless otherwise noted. Antibodies used were mouse monoclonal anti-yeast Ras2 (Santa Cruz Biotechnology catalog number sc-365773; 3% milk), mouse monoclonal anti-Pan Ras (Cytoskeleton catalog number AESA02), mouse monoclonal anti-GFP (ThermoFisher catalog number MA5-15256), rabbit polyclonal anti-Kar2, sheep polyclonal HRP-anti-mouse (GE Healthcare catalog number NA931), goat polyclonal HRP-anti-mouse (Kindle Biosciences catalog number R1005), and goat polyclonal HRP-anti-rabbit (Kindle Biosciences catalog number R1006) (58).

**5-FOA plasmid-loss assay.** The qualitative 5-FOA plasmid-loss assay was performed based on a previously described method and further adapted to obtain quantitative results (17). For the qualitative assay, yeast grown on SC-Ura, Leu solid medium were patched onto YPD solid medium and incubated for 2 to 3 days. The resultant patches were replica plated onto 5-FOA solid medium and incubated for 2 to 4 days. YPD and 5-FOA plates were imaged with a Canon flat-bed scanner (300 dpi), and images were equivalently processed using Adobe Photoshop for minor rotation, crop, and contrast adjustment. For the quantitative assay, the yeast described above were cultured in SC-Ura, Leu liquid medium for 2 days ($A_{600}$ of ~2 to 3) and diluted with fresh medium, and a portion of the diluted culture estimated to have ~300 CFU was plated onto YPD solid medium and incubated for 2 to 3 days. Colonies on YPD solid medium were counted and then replica plated onto 5-FOA solid medium, and colonies on 5-FOA medium were counted after 2 to 4 days. For each condition, the ratio of colony numbers on 5-FOA medium to those on YPD was calculated and expressed as a percentage relative to the positive-control Ras2-CIIS (i.e., 100% reference condition).

**Acyl-PEG exchange.** The acyl-PEG exchange assay was based on previously described methods (29). Yeast cultured in SC-Leu medium ($A^{600}$, ~0.9 to 1.0) were lysed by mechanical agitation with silica beads (i.e., bead-beating), and equivalent portions of total protein from each lysate (200 $\mu$g) were subjected to methanol-chloroform precipitation. The protein pellet was dried by speed-vac, resuspended in TEA buffer (50 mM triethanolamine [pH 7.3], 150 mM NaCl) containing 4% SDS and 4% EDTA (TEA/S/E), and treated with hydroxylamine (HAM; 1 M final) and Triton X-100 (0.15% final) in TEA buffer. The control sample (−HAM) was mock treated with an equivalent volume of TEA buffer containing 0.15% Triton X-100. After a 2-h nutation, protein samples were precipitated, resuspended in TEA/S/E, and treated with 5-kDa mPEG-Mal (1.5 mM final) and

Triton X-100 (0.15% final) in TEA buffer. After a 3-h nutation at room temperature, protein was recovered by methanol-chloroform precipitation, resuspended in 1× Laemmli sample buffer, heated (95°C, 7 min), and analyzed by 15% SDS-PAGE and immunoblotting.

**Cycloheximide chase.** A previously reported method was adapted for this study (59). Cycloheximide-treated cultures were monitored by $A_{600}$ to confirm growth arrest. In brief, yeast were cultured in SC-Leu medium at 29°C to an $A_{600}$ of 1.2 to 1.9 (calculated by measuring $A_{600}$ of an 11-fold dilution). Cells were harvested by centrifugation, resuspended to $A_{600}$ of 1.5 in fresh medium, and incubated for 1 h at 29°C. Cycloheximide was then added to the cultures (250 $\mu$g/mL), and the first time point sample ($t = 0$ min) was immediately collected (950 $\mu$L) and mixed with ice cold "stop mix" (50 $\mu$L; 0.2 M sodium azide, 5 mg/mL bovine serum albumin). Additional time point samples (1 to 26 h) were collected in a similar manner. Cells from each time point sample were harvested by centrifugation, whole-cell lysates prepared by alkaline lysis, and the lysates were analyzed by SDS-PAGE and immunoblotting using anti-Ras2 and anti-Kar2 antibodies. Antibody complexes were detected by chemiluminescence, and images were digitally captured as described above. Immunoblot band intensities from digital images were evaluated using ImageJ and are reported as the amount relative to that observed for the initial time point (i.e., $t = 0$ min). Ras2 levels were quantified using at least 2 exposures for each Ras2-CAAX variant. Kar2 levels were quantified using the exposures from all five Ras2-CAAX variants.

**Ras pulldown activation assay.** This Ras pulldown was performed using the Ras Pulldown Activation Assay Biochem kit (Cytoskeleton, Inc.) based on the manufacturer's instructions and prior methods (33). Yeast cultured in SC-Leu (5-FOA selected) or SC-Ura, Leu (not 5-FOA selected) liquid medium were lysed by bead-beating using kit-provided lysis buffer and protease inhibitors, and lysates were stored at −80°C until needed. The lysates were diluted with lysis buffer to the same protein concentration, and equal portions of each lysate were reserved as the input sample while another portion (300 $\mu$g) was incubated with 30 $\mu$L kit-provided GST-tagged Raf1-RBD-coated beads for 1 h with nutation, followed by washing of bead-treated samples according to kit instructions. The samples were resuspended in 2× Laemmli sample buffer and heated (95°C, 5 min), and equal volumes of each sample were analyzed by SDS-PAGE followed by immunoblotting with kit-provided anti-Pan Ras antibody. Activated Ras2 levels were quantified from the immunoblot using ImageJ by calculating the ratio of Ras2 pulled-down with beads (i.e., Ras2-GTP) to input Ras2 (i.e., total Ras2).

**Differential fractionation.** Yeast cultured in SC-Ura, Leu liquid medium ($A_{600}$, ~0.9 to 1.0) were lysed by mechanical agitation with silica beads (4 min beating, 2 min on ice, 4 cycles) into TSE buffer (50 mM Tris [pH 7.5], 200 mM sorbitol, 1 mM EDTA, with protease inhibitor 1 mM phenylmethylsulfonyl fluoride, and 1 $\mu$g/mL each of aprotinin, chymostatin, leupeptin, and pepstatin). The lysate was clarified (500 × $g$, 10 min, 4°C), half of the clarified lysate was saved as the input sample (i.e., total), and the remainder was fractionated (100,000 × $g$, 1 h, 4°C). The supernatant (S) was transferred into a new tube, and the pellet (P) was washed twice with TSE buffer before resuspending to the original sample volume used for fractionation. The total, S and P fractions were mixed with 6× Laemmli sample buffer prior to analysis by SDS-PAGE and immunoblotting.

**Yeast GFP microscopy.** Yeast cultured in SC-Ura liquid medium ($A_{600}$, ~0.9 to 1.0) were spotted onto positively charged glass slides (Globe Scientific Inc.) and imaged by fluorescence microscopy with a Zeiss Axio Observer Z1 inverted microscope equipped with fluorescence optics and Plan-Apochromat 63× 1.4 numerical aperture objective. Representative images were captured using AxioVision software, typically with exposures of 800 to 1,000 ms.

***ras2-23*(TS) temperature sensitivity assay.** The temperature sensitivity assay was adapted from a previously described method (36). Yeast patches cultured on SC-Ura solid medium at room temperature for 3 to 4 days were replica plated onto two SC-Ura plates for incubation at either room temperature or 37°C for 3 to 5 days. Plates were imaged and processed as described for the 5-FOA plasmid-loss assay.

**Acute heat shock assay.** The acute heat shock assay was adapted from a previously described method (17). Yeast cultured in SC-Leu liquid medium for 3 days ($A_{600}$, ~2.5 to 4) were diluted with fresh medium to have matching densities, and equally apportioned volumes of normalized cultures were transferred into two microcentrifuge tubes: one for acute heat shock (55°C, 10 min) and a second for mock treatment (i.e., no heat). Treated and mock-treated samples were transferred to a 96-well plate and subjected to 5-fold serial dilution with fresh medium, and the dilution series was pinned onto SC-Leu and YPD solid media, respectively. In parallel, appropriate dilutions of cells were spread on SC-Leu solid medium for colony counting. After incubation for 3 to 4 days at 30°C, pinned plates were imaged and processed as described for the 5-FOA plasmid-loss assay, and colonies were counted.

***cdc25*(TS) temperature sensitivity assay.** The temperature sensitivity assay was adapted from previous studies (37, 38). Yeast were cultured in SC-Leu liquid medium to saturation for 3 days at room temperature, normalized to equivalent density, and subject to 10-fold serial dilution. The dilutions were pinned onto two SC-Leu plates; one was incubated at room temperature and the other at 37°C for up to 6 days. Plates were imaged and processed as described above for the 5-FOA plasmid-loss assay.

## SUPPLEMENTAL MATERIAL

Supplemental material is available online only.

**SUPPLEMENTAL FILE 1**, PDF file, 3.8 MB.

## ACKNOWLEDGMENTS

We thank Robert Deschenes (USF Health) for early discussions about this work and for providing yeast strains (LRB938 and RJY510) and plasmids (B250, B561, B562, Ras2-Ssaax, and Ras2-Scaax), Ami Aronheim (Technion-Israel Institute of Technology) for the *cdc25-2* yeast strain, Jasper Rine (UC Berkeley) for the JRY5318 yeast strain, Jeff Brodsky (University of Pittsburgh) for

rabbit polyclonal anti-Kar2 antibody, and Schmidt lab members for constructive feedback and technical assistance with plasmid constructions and experiments.

This research was supported by Public Health Service grant GM132606 from the National Institute of General Medical Sciences (W.K.S.) and funds from New York University Abu Dhabi (T.M.D.).

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
