## [Reviewer comments · Microbiology Spectrum]

Microbiology Spectrum

Specific disruption of Ras2 CAAX proteolysis alters its localization and function

Rajani Ravishankar, Emily Hildebrandt, Grace Greenway, Nadeem Asad, Sangram Gore, Timothy Dore, and Walter Schmidt

Corresponding Author(s): Walter Schmidt, The University of Georgia

Review Timeline:

Submission Date:	July 14, 2022
Editorial Decision:	August 17, 2022
Revision Received:	November 30, 2022
Accepted:	December 6, 2022

Editor: Robert Arkowitz

Reviewer(s): Disclosure of reviewer identity is with reference to reviewer comments included in decision letter(s). The following individuals involved in review of your submission have agreed to reveal their identity: Jarrod R. Fortwendel (Reviewer #1)

Transaction Report:

DOI: <https://doi.org/10.1128/spectrum.02692-22>

August 17, 2022

Dr. Walter K Schmidt
University of Georgia
Athens

Re: Spectrum02692-22 (Specific disruption of Ras2 CAAX proteolysis alters its localization and function)

Dear Dr. Walter K Schmidt:

The two reviewers both found this work to be solid, clear and well-written and it details a substantial advance that will be of broad interest. Both reviewers had some specific comments with regard to expression levels which should be addressed. Furthermore reviewer #2 also indicated that the data in Fig. 4B and Fig 5A appears to be based upon a single experiment and this should be directly addressed.

Link Not Available

Sincerely,

Robert Arkowitz

Journals Department
Reviewer comments:

Reviewer #1 (Comments for the Author):

This clear, well-written manuscript describes the use of a unique CAAX motif to study the effects of specific inhibition of Ras2 post-translational proteolysis in *Saccharomyces cerevisiae*. The authors use a range of well established, Ras protein-specific assays to build upon prior findings (in yeast and other organisms) to further our understanding of Ras protein PTMs and how they impact Ras signaling. Although simple on its face, this study is unique in that previous attempts to do so have utilized mutant strains that potentially have widespread defects in protein maturation. These studies, for the first time, focus on specific inhibition of Ras processing. The studies here lay a foundation for future work to specifically analyze additional CAAX motif-bearing proteins and will help us move toward a better understanding of how Ras protein maturation is regulated. Overall, the

manuscript is clear and the conclusions are supported by the data.

1. The only substantive suggestion I have is that the data showing changes in protein abundance that play a very central role in the manuscript would be more strongly supported by the addition of complementary gene expression data. Simple RT-qPCR experiments would suffice to show that all mutant strains express Ras alleles at the same transcriptional level. Protein abundance could then securely be interpreted as transcription-independent.

Reviewer #2 (Comments for the Author):

The manuscript "Specific disruption of Ras2 CAAX proteolysis alters its localization and function" by Ravishankar et al, introduces a novel and useful method to investigate the function of CAAX proteolysis, which is applicable to a wide range of CAAX proteins, and, in my opinion, to most model organisms. This represents a substantial advance over previous methods that likely resulted in pleiotropic defects and hence corresponding difficulties in interpretation. In a previous study, authors identified an atypical CAAX protein (Ydj1) which only undergoes prenylation. By using the atypical sequences, they were able to investigate the impact of altering the CAAX processing dynamics (farnesylation, proteolysis and palmitoylation) in a well-studied protein, the GTPase Ras2 of budding yeast. Authors showed that Ras2 farnesylation is required to support viability and then demonstrate that the absence of proteolysis did not affect the palmitoylation of Ras2. To explain the reduced viability of some of the forms (the only-farnesylated; SCASQ, and the not processed; SSIIS), they quantified the amount of total protein present and of activated GTPase, to confirm that the inability of these Ras2-CAAX forms is not due to low protein or GTP activation levels. Further, authors confirmed that the absence of proteolysis/palmitoylation or directly the absent of full processing, negatively affected the presence and stability of these Ras2-CAAX forms at the plasma membrane, and therefore their functionality. Interestingly, the authors showed that the reduced activity of these isoforms was also observed in the hyperactive form of Ras2 (Ras-2-G19V), clearly showing that the impact of this hyperactivation was reduced in the Ras2-CAAX without proteolysis/palmitoylation or with the absence of full processing. All together indicating for the first time that at least 2 PTM are needed for the full activity and for the normal protein abundance of Ras2 in budding yeast.

The work is very quantitative, and the majority of conclusions are based on more than one experimental approach, combining biochemistry and cell imaging, as for example in figure 5. The only negative aspect is that, as state in some figure legends (i.e., Fig 4B or Fig. 5A), some conclusions are based on a single experiment for some of the Ras2-CAAX sequences. I completely understand that time is an issue, but I strongly suggest the authors to carry out 2-3 biological replicates that are missing.

There are some small aspects of the result section that could be clarified. Expression levels (transcriptional) of all variants should, in principle, be similar. However, as stated in lane 312, proteins levels were higher for all Ras2-CAAX constructs, being the highest in SCASQ and SSIIS, the former farnesylated only and the latter lacking any processing. The authors discuss the possibility of a compensatory effect, due to the requires for viability. The main difference between CIIS, SCIIS, CASQ with SCASQ and SSIIS is that the first three are most targeted to and stable at the plasma membrane/internal membranes, while the latter two are not (as shown in Figure 5). This result is intriguing, as some protein extraction methods are not optimal for purification of plasma membrane proteins. Is it possible that the extraction method used biases some of the findings presented? Please comment on this. Additionally, given that all images were taken with the same settings, is it, for example, possible to quantify the total signal detected in Figure 5 to compare the values per cell (total amount of Ras2-CAAX-GFP) between the different CAAX domains? In principle, similar protein abundance should be obtained that the shown at the western blots.

To sum up, this is an elegant and concise study, very well organized scientifically and clearly written that was a pleasure to read. As a minor comment, it is somewhat difficult to follow how each CAAX modification was affected (although Figure 1B is useful). I suggest the authors an alternative name scheme for each PTM within the main text and figures to facilitate the flow of the manuscript.

Staff Comments:

Preparing Revision Guidelines

- Point-by-point responses to the issues raised by the reviewers in a file named "Response to Reviewers," NOT IN YOUR COVER LETTER.
- Upload a compare copy of the manuscript (without figures) as a "Marked-Up Manuscript" file.
- Each figure must be uploaded as a separate file, and any multipanel figures must be assembled into one file.
- Manuscript: A .DOC version of the revised manuscript

- Figures: Editable, high-resolution, individual figure files are required at revision, TIFF or EPS files are preferred

Please return the manuscript within 60 days; if you cannot complete the modification within this time period, please contact me. If you do not wish to modify the manuscript and prefer to submit it to another journal, please notify me of your decision immediately so that the manuscript may be formally withdrawn from consideration by Microbiology Spectrum.

Response to Reviewers

Reviewer #1

1) ... the data showing changes in protein abundance that play a very central role in the manuscript would be more strongly supported by the addition of complementary gene expression data. Simple RT-qPCR experiments would suffice to show that all mutant strains express Ras alleles at the same transcriptional level. Protein abundance could then securely be interpreted as transcription-independent.

Response: New data clearly demonstrates that different protein half-lives contribute to the observed differences in protein levels of the Ras2-CAAX variants evaluated (see Fig. 4B and Fig. S3). The mechanism that regulates Ras2 half-life remains unclear but will be investigated in the future.

Reviewer #2

1) ... in some figure legends (i.e., Fig 4B or Fig. 5A), some conclusions are based on a single experiment for some of the Ras2-CAAX sequences. I completely understand that time is an issue, but I strongly suggest the authors to carry out 2-3 biological replicates that are missing.

Response: The replicate issue is with one sample in each of the indicated figures. For Fig. 5A, we have completed replicates for Ras2-SCASQ and have updated the results, which remain consistent with prior findings. For Fig. 4B (now Fig. 4C), the replicates of other samples fully reproduced in replicate experiments, so we have confidence in the methodology and the results of the singleton Ras2-SSIIS sample. Another mitigating factor is that we had exhausted the costly Ras Pull-down kit needed for generating the replicate (>\$800 USD), and the cost of procuring a new kit was too excessive for our limited budget. We are willing to modify the impacted figure to remove the singleton sample, but it is our opinion that doing so will yield a less comprehensive result that will likely lead to reader confusion.

2) Is it possible that the extraction method used biases some of the findings presented? Please comment on this.

Response: The method used to obtain lysates is considered by the yeast community to be the optimal method for obtaining unbiased whole cell protein extracts. An alternative method that involves mechanical cell breakage offers no improvement over the method reported in our study, and if not performed carefully is more susceptible to protein breakdown during sample processing. Even so, this alternative method was employed to validate protein level differences (e.g., see Fig. S2C), and similar results were observed.

3) ... is it, for example, possible to quantify the total signal detected in Figure 5 to compare the values per cell (total amount of Ras2-CAAX-GFP) between the different CAAX domains?

Response: Rather than trying to quantify the total GFP signal from cells as an indirect measure of GFP-Ras2 protein levels, we directly evaluated protein levels by immunoblot to be consistent with other figures. The protein levels of GFP-Ras2-CAAX variants are now reported as a new panel (i.e., Fig. S4A), and the results indicate similar protein levels for GFP-tagged Ras2-CAAX variants with perhaps GFP-Ras2-SSIIS being slightly elevated relative to other variants. We interpret these results to indicate either that the tag interferes with the turnover of GFP-tagged Ras-CAAX variants or that their overexpression interferes with the normal turnover process of Ras2.

4) As a minor comment, it is somewhat difficult to follow how each CAAX modification was affected (although Figure 1B is useful). I suggest the authors an alternative name scheme for each PTM within the main text and figures to facilitate the flow of the manuscript.

Response: We have attempted to use alternative naming schemes in the past for describing the various PTM combinations expected for each CAAX variant, mostly in an effort to simplify language and be more concise, but this has often led to even more reader confusion. Hence, we have opted to label samples by their CAAX motifs in our study in order to be as precise as possible. The purpose of Fig. 1B, as noted by the reviewer, is intended to help navigate the reader through the complexity of PTMs associated with each CAAX variant.

December 6, 2022

Dr. Walter K Schmidt
The University of Georgia
Athens

Re: Spectrum02692-22R1 (Specific disruption of Ras2 CAAX proteolysis alters its localization and function)

Dear Dr. Walter K Schmidt:

Thank you for completely addressing the points raised by both reviewers.

Your manuscript has been accepted, and I am forwarding it to the ASM Journals Department for publication. You will be notified when your proofs are ready to be viewed.

Sincerely,

Robert Arkowitz
Editor, Microbiology Spectrum

Journals Department
Supplemental Material for Publication: Accept